# Anthelmintic Activity of *Tanacetum vulgare* L. (Leaf and Flower) Extracts against Trichostrongylidae Nematodes in Sheep In Vitro

**DOI:** 10.3390/ani13132176

**Published:** 2023-07-02

**Authors:** Alīna Kļaviņa, Dace Keidāne, Kristīne Ganola, Ivars Lūsis, Renāte Šukele, Dace Bandere, Liga Kovalcuka

**Affiliations:** 1Institute of Food and Environmental Hygiene, Faculty of Veterinary Medicine, Latvia University of Life Sciences and Technologies, K. Helmana Street 8, LV-3004 Jelgava, Latvia; dace.keidane@lbtu.lv (D.K.); kristine.ganola@lbtu.lv (K.G.); ivars.lusis@lbtu.lv (I.L.); 2Department of Pharmaceutical Chemistry, Faculty of Pharmacy, Rīga Stradiņš University, Konsula Street 21, LV-1007 Riga, Latvia; renate.sukele@rcmc.lv (R.Š.); dace.bandere@rsu.lv (D.B.); 3Department of Pharmaceuticals, Red Cross Medical College of Rīga Stradiņš University, J. Asara Street 5, LV-1007 Riga, Latvia; 4Baltic Biomaterials Centre of Excellence, Headquarters at Riga Technical University, Dzirciema Street 16, LV-1007 Riga, Latvia; 5Clinical Institute, Faculty of Veterinary Medicine, Latvia University of Life Sciences and Technologies, K. Helmana Street 8, LV-3004 Jelgava, Latvia; liga.kovalcuka@lbtu.lv

**Keywords:** phytotherapy, plant extracts, *Tanacetum vulgare*, sheep, gastrointestinal parasites, anthelmintic resistance, egg hatching test, larval development test

## Abstract

**Simple Summary:**

Different types of endoparasites are present in small ruminants, especially in sheep. One of the most important parasites are the gastrointestinal nematodes (Trichostrongylidae). Emaciation, anorexia, decreased productivity, and diarrhea are observed in infected animals. These infections cause great economic damage to sheep owners through production losses and treatment costs. In connection with the frequent treatment of sheep against endoparasite infections, anthelmintic resistance has developed. Due to the global anthelmintic resistance situation, the principles of parasite control have changed. New alternatives have been sought, and one of the most researched alternatives is phytotherapy. Different plants with anthelmintic activity would be valuable for the control of ruminant helminths. Phytotherapy would reduce the use of synthetic drugs and reduce treatment costs.

**Abstract:**

Due to the high prevalence of gastrointestinal nematodes in sheep, the growing anthelmintic resistance, and the development of organic farming systems, sustainable alternatives are being sought. One such method is phytotherapy. This study aimed to evaluate the in vitro ovicidal and larvicidal activity of extracts of tansy (*Tanacetum vulgare* L.) growing in Latvia on gastrointestinal nematodes (Trichostrongylidae) in sheep. The leaves and flowers of the tansy were extracted separately in 70%, 50%, and 30% ethanol and acetone. Six concentrations were prepared from each extract 500 mg/mL, 200 mg/mL, 100 mg/mL, 50 mg/mL, 20 mg/mL, and 10 mg/mL. In vitro egg hatching test and micro-agar larval development test were performed. Extracts of tansy have strong larvicidal activity. The highest percentage of larvae inhibition for most of the extracts was 100%, but for egg inhibition, it was 95.8% for the 200 mg/mL concentration of 50% acetone and 93.3% for the 500 mg/mL concentration of 50% ethanol leaf extracts. All tansy extracts had ovicidal and larvicidal activity against Trichostrongylidae in sheep.

## 1. Introduction

Some of the most common sheep gastrointestinal parasites are among the Trichostrongylidae nematodes family (*Haemonchus contortus*, *Ostertagia* spp. *Trichostrongylus* spp., *Cooperia* spp.). They all belong to the order Strongylida. Emaciation, anorexia, and decreased productivity are observed in infected sheep [1]. Diarrhea is a more characteristic clinical sign in Ostertagiosis and Trichostrongylatosis [2]. On the other hand, the clinical signs of Haemonchosis are anemia and submandibular edema (“bottle jaw”) [2]. Moreover, in young animals, the clinical signs are often more severe and significant. These infections cause great economic damage to the sheep owners, with a total loss amount of up to EUR 1.8 billion reported in European countries [3]. These losses comprise lost production (wool, meat, milk) and high costs of animal treatment, specifically the purchase and application of deworming drugs [3].

In general, the range of deworming drugs available for sheep is not that wide. Most often, they are broad-spectrum. The most commonly used groups are macrocyclic lactones (Ivermectin), benzimidazoles (Albendazole), imidazothiazoles (Levamisole), and amino-acetonitrile-derivates (Monepantel). Until 2000, there were few reports of anthelmintic resistance, but since 2000, the number of reports on the resistance to various anthelmintics has increased rapidly, especially in small ruminants. There are reports from different countries around the world and this is also an actual problem in Europe. There are reports both from southern countries such as Spain, Greece, France, and Italy and also from northern countries such as Denmark, Sweden, and Norway [4,5,6] as well as Latvia, where resistance to Ivermectin and Albendazole was detected [7]. Multiple resistance has also been reported in some countries [6].

Due to the high prevalence of gastrointestinal nematodes, the growing anthelmintic resistance today, and the development of organic farming systems, the perception of parasite control has changed worldwide. Today, there is more emphasis on sustainable parasite control, which includes pasture management, genetics, nutrition, and other sustainable alternatives [8]. One of these sustainable alternative methods is reported phytotherapy. By using an effective alternative, it would be possible to reduce the frequency of use of chemical anthelmintics and reduce the development of anthelmintic resistance. This would especially benefit organic farms [9]. In organic farms, parasite control is very important because they have restrictions for chemical drug usage and a double withdrawal period [10]. In addition, phytotherapy research serves as a source of new anthelmintic drugs [3] and herbal remedies have a much lower chance of resistance [8].

Latvia has rich flora with various plants, including those that may have anthelmintic activity. In Latvian folk medicine, antiparasitic properties are often attributed to Tanacetum family plants, especially tansy (*Tanacetum vulgare* L.) [11,12]. It is a widespread perennial plant in Latvia, which often grows on roadsides, meadows, and river banks. It is a relatively large plant, 40–140 cm in height with a specific odor. In Latvia, it blooms from July to October. In total, there are four plants of the Tanacetum genus in Latvia—*Tanacetum balsamita* L., *Tanacetum corymbosum* L., *Tanacetum parthenium* L., and *Tanacetum vulgare* L. However, only tansy is described in folk medicine and used for parasite control. Moreover, in Scandinavian countries, it is often used on organic farms as a parasite preventative [13]. They cultivate tansy in animal pastures and others that have pasture rotation with paddocks, making a specific deworming paddock with tansy [13]. Plants contain different bioactive compounds, such as saponins, alkaloids, non-protein amino acids, polyphenols (tannins), lignin, and glycosides [14]. Their mode of pharmacological action can be different. It has been reported that tansy not only has an antiparasitic effect, but it also has other effects, such as antioxidant, anti-inflammatory, antimicrobial, and cytotoxic [15,16,17].

In general, there is a lack of knowledge about the antiparasitic effects of specific plants in the scientific literature on animals, especially ruminants. Therefore, this study aimed to evaluate the anthelmintic activity of tansy in vitro on sheep gastrointestinal nematodes (Trichostrongylidae) using the egg hatching test (EHT) and the larval development test (LDT).

## 2. Materials and Methods

### 2.1. Ethical Approval

According to the legislation, the procedures conducted in this study did not require ethical approval. Parasitic eggs and larvae were obtained from fecal samples that were taken during the routine animal examination and sent to the Latvia University of Life Sciences and Technologies Faculty of Veterinary Medicine Institute Food and Environmental Hygiene Parasitology laboratory. Animal examination and sample collection were performed by a certified veterinarian and the procedure did not exceed principles of good veterinary practice, and animal welfare and was not painful.

### 2.2. Plant Material Preparation

The aerial parts of the tansy were harvested during flowering in July 2020, Allaži parish (57.083579, 24.782287), Latvia. The plant materials were identified, and the leaves and flowers were separated by hand and dried in the shade at an ambient temperature. Reference samples were kept in the internal collection of the Rīga Stradiņš University Pharmaceutical Chemistry department, with the labels: BZ-2020 and BL-2020. The extracts were prepared by Rīga Stradiņš University, Faculty of Pharmacy, Department of Pharmaceutical Chemistry, by adding 10 g of powdered plant materials and 100 mL acetonic (Chempur, Poland) or ethanolic solution (Kalsnavas Elevators, Latvia) at three concentrations 70%, 50%, and 30%. Water was purified using the Stakpure GmpH Water System (Niederahr, Germany). The extracts were macerated for 80 min using an orbital shaker (PSU-10i Biosan, Latvia) and filtered through 9 mm filter papers (Sartorius, FT-3-303-110, Hamburg, Germany), and the solvent was removed via rotary evaporation (Heidolph Laborota 4002 control, Schwabach, Germany) at 55–60 °C 2 h. The final result was 5 mL of six different semi-dry extracts, which were frozen at the temperature of −20 °C until analysis. Six concentrations were prepared from each extract type 500 mg/mL, 200 mg/mL, 100 mg/mL, 50 mg/mL, 20 mg/mL, and 10 mg/mL. Overall, 72 different concentration extracts were obtained. All procedures were carried out according to the general WHO guidelines for plant collection and processing [18].

### 2.3. In Vitro Anthelmintic Activity

#### 2.3.1. Parasites and Recovery/Obtaining of Eggs

Fecal samples were collected rectally from sheep naturally infected with Trichostrongylidae, during routine animal examination. Samples were transported to the laboratory in transport vials at the temperature of 4 ± 2 °C on the same day within 4–6 h for further examination. Parasite infection was confirmed by the modified McMaster method (eggs per gram of feces); 4 g of feces and 56 mL salt solution, and an egg suspension was prepared from the selected samples according to Coles et al. [19]. First, 20 g of feces was crushed in 200 mL of distilled water and was filtered through sieves (1000 µm, 100 µm, 50 µm, and 32 µm). The residue of the last sieve was collected with distilled water and was centrifuged (LMC-3000, Biosan, Latvia) for 10 min at 2000 rpm. The supernatant was discarded, and a salt solution (NaCl with a density of 1.018 g/mL) was added. The mixture was centrifuged for 15 min at 3000 rpm, and the supernatant was sifted through a 32 µm sieve. The eggs were collected from the sieve with distilled water.

#### 2.3.2. Egg Hatching Test

The test was performed according to the World Association for the Advancement of Veterinary Parasitology (WAAVP) recommendations, as described by Coles et al. [20]. Approximately 100–200 eggs were added in each well of 72-well plates at a total volume of 350 µL. There, the same volume was added to each extract concentration. Positive Ivermectin (Biomectin 1%, Vetoquinol Biowet, Gorzów Wielkopolski, Poland), and negative distilled water, control wells were created on the plate. Three replicates were performed for each dilution. The plate was incubated for 48 h at +25 °C; then, 5 µL of Lugol’s iodine solution (10 mg/g) was added to stop egg hatching. Non-embryonated and embryonated eggs and first-stage larvae (L_1_) were counted to calculate the inhibition of eggs (%).

#### 2.3.3. Larval Development Test

The micro-agar larval development test (MALDT) was performed according to Coles et al. [19]. The egg suspension preparation was described in Section 2.3.1. We added 150 µL of 2% Agar Bacteriological No.1 (Oxoid™, Brno, Czech Republic) at 45 °C in 72-well plates. After the agar was solidified, a 150 µL egg suspension was added (approximately 50 eggs per well), which was diluted with a solution of Amphotericin B (Sigma-Aldrich^®^, St. Louis, MO, USA) (1:1). We added 150 µL of yeast extract in each well, prepared as described by Hubert and Kerboeuf [21] (1 g yeast extract (Sigma-Aldrich^®^, St. Louis, MO, USA) plus 90 mL 0.85% NaCl, autoclaved for 20 min, and 3 mL of 10 × concentrated Earl’s solution (Sigma-Aldrich^®^) per 27 mL of yeast extract added) and the same volume of each extract concentration. The positive control was Ivermectin (Biomectin 1%, Gorzów Wielkopolski, Vetoquinol Biowet, Poland), and the negative control was distilled water. Three replications were performed for each dilution. Distilled water was added to the empty wells to prevent evaporation. The plate was incubated for 7 days at +25 °C. After incubation, 5 µL of Lugol’s iodine solution (10 mg/g) was added, and the eggs (non-embryonated and embryonated) and larvae (L_1_, L_2,_ and L_3_) were counted.

### 2.4. Statistical Analysis of Data

The inhibition of egg hatching (%) was calculated using the formula I_eggs_ = 100 × (1 − P_test_/P_non-treated_), P-number of eggs hatched [16]. The inhibition of larvae (%) was calculated using the formula I_larvae_ = 100 × (L_1_ + L_2_ + eggs/L_1_ + L_2_ + L_3_ + eggs) [22]. The percent of inhibition was calculated using Microsoft Excel 2016. The effects of each plant extract type on the eggs and larvae were analyzed by STATA 17.0 statistical software, using linear regression (I_eggs_) and logistic regression (I_larvae_ > 90%). To calculate the median lethal concentration (LC_50_) for each tested plant extract type, the data were transformed to the logarithm (Log10) and analyzed by Probit analysis [23].

## 3. Results

### 3.1. Ovicidal Activity

The tansy ethanol and acetone extract from leaves Table 1 and flowers Table 2 had varied effects on the Trichostrongylidae eggs at all concentrations.

The plant part was a significant factor (*p* = 0.002) when considered separately. The leaf extract was more effective than the flower extract. Although, when the part of the plant was compared together with other factors, their significance decreased (*p* = 0.235). This was the same with the extract’s solvent. However, the effect of acetone was more significant (*p* = 0.011) than that of ethanol. The interaction between the plant part and solvent was essential (*p* = 0.036); the best combination was the ethanol extract of the tansy leaves. The solvent’s concentration was an important factor. A solvent concentration of 50% showed a significantly different result than 30% or 70%. The order of concentration was observed from 100 mg/mL to 500 mg/mL, except for the 50% solvent concentration.

The highest percentage of egg inhibition from the tansy leaf extract was with the 200 mg/mL concentration of 50% acetone extract (95.8%), the 500 mg/mL concentration of 50% ethanol extract (93.3%), and the 500 mg/mL concentration of 50% acetone extract (92.3%) Table 1.

The highest percentage of egg inhibition from the tansy flower extract was with the 500 mg/mL and 100 mg/mL concentrations of the 50% acetone extracts (91.5% and 84.3%) Table 2.

The leaf extracts had lower LC_50_ values than the flower extracts. The lowest LC_50_ value of 0.04 mg/mL was for the 30% acetone extract of tansy leaves Table 1.

### 3.2. Larval Inhibition Activity

All tansy leaf Table 3 and flower Table 4 extracts had larvicidal effects on Trichostrongylidae larvae.

Similar to the previous results, the leaf extracts showed stronger larvicidal activity (*p* = 0.005). There were also differences in the concentrations of the extract solvent. The 30% extracts had better results on larvae than the 50% or 70% extracts. The solvent and dilution variation did not cause significant differences (*p* = 0.580 and *p* = 0.507). Almost all extracts had similar effectiveness at 100%. The lowest larvicidal activity was the flower 30% acetone extracts and the 10 mg/mL concentration of the acetone extracts Table 3.

The lowest LC50 value of 0.005 mg/mL was for the 70% acetone extract of tansy flowers Table 4.

## 4. Discussion

One of the plants with antiparasitic activity in folk medicine is tansy. It is a plant from the Asteraceae family and several of its members have been described to have antiparasitic effects. These properties are related to plant secondary metabolites, which are primarily necessary for plant protection [14]. In general, plants have three groups of these metabolites: terpenes, phenolic compounds, and nitrogen-containing compounds [24]. These compounds are formed in different parts of plants, such as the root, stem, leaves, fruits, and flowers, and their concentration in these parts can vary [25]. The variation depends on many different factors: soil, climate, season, plant development stage, and others [26,27]. Each plant also contains a different combination of these components. They can have synergistic or antagonistic action [26]; for example, polyvinyl polypyrrolidone is an inhibitor of polyphenols [14].

Significant differences in concentration between fresh plants and their extracts were observed. An important factor, in this case, is the extraction process [26], especially the choice of solvent [28]. Two solvents were used in this study ethanol and acetone. They are most commonly used to prepare plant extracts, and both are polar solvents that best extract the plant’s polar components, such as phenolics [29]. Due to the changing chemical composition of the plant, it is recommended to carry out studies using plants growing in a given country and determining their chemical composition. In this study we used tansy which was growing in Latvia.

Asteraceae family plants contain more terpenes (sesquiterpene lactones and santonin [13]) and phenolics (phenolic acids and lignans [30]). Tansy from Latvia contains different phenolic compounds; in a study from Latvia, chemical compounds in extracts of tansy were tested, where total flavonoids, phenolic acids, total phenols, tannins, specific phenols (quinic acid, gallic acids, catechin, chlorogenic acid, caffeic acid, rutin, quercetin, kaempferol, and apigenin) and essential oils (thujone) was detected [31]. Overall, phenolic compounds are the secondary metabolites, which are often related to antiparasitic effect. The leaf extracts (135–219 mg GAE/g) contain more phenolic compounds than flower extracts (127–155 mg GAE/g) [31]. This could be the reason why in this study leaf extracts showed stronger anthelmintic activity than flower extracts. However, overall acetone and ethanol extracts contained phenolics and that is why both of them still have good ovicidal and larvicidal effects. Besides larval development, the test results were better than the egg-hatching test. There are relatively few studies explaining the mechanism of action of phenols on parasite`s eggs [32], but more information is available on the action on the parasite itself, which causes parasite death by interfering with its energy metabolism and causing damage to cuticle cells [33,34].

Phenolics are an important part of phytotherapy because they have the most anthelmintic properties, especially tannins [14]. Extracts of *T.vulgare* growing in Latvia contain tannins and their quantity is higher in the leaf extract (10–19%) than in the flower extract (3–5%) [31]. There are two types of tannins: hydrolyzed and non-hydrolyzed or condensed. Condensed tannins are the most investigated substation in research worldwide. Two variations still exist regarding the effect of tannin on parasites. First, they form complexes with proteins, thus interfering with the feeding and movement function of the parasite; second, it has a positive effect on the animal`s immune system [24,27,35]. Other positive effects of using plants containing condensed tannins in animals are often described; they act on production quality and quantity (wool, milk, meat), and reproductions [36]. The effect of tannins depends on the animal species. There is a difference between ruminants and monogastric species [37]. Additionally, not all ruminants can tolerate tannins equally. More affected by tannin are sheep and cattle than goats and deer [36]. Goats are less affected because they have the tannin-degrading bacteria *Streptococcus caprinus* in their rumen, higher urea recycling, and constant secretion of protein-rich saliva [38]. Unlike sheep, protein digestion in the rumen is not affected. This also appeared in an in vivo study where the effect of *T.vulgare* extract was evaluated on gastrointestinal nematodes; the intensity efficiency was lower for goats (34.2%) than for sheep (63.7%) [39].

On the other hand, other studies indicate that thujone also has antiparasitic properties [15,40]. Thujone (monoterpene ketone) is the main chemical component of tansy [13]. There are two types of thujone α-thujone and β-thujone. In different *Tanacetum* genus plants, the form and concentration of thujone can vary greatly. For example, *T.vulgare* contains more β-thujone, but *T. argyrophyllum* contains more α-thujone [41,42]. The concentration varies in the part of the plant; the flowers had a slightly higher concentration than the leaves [41]. A similar result was also obtained for the tansy growing in Latvia; the flowers contained 0.67–5.98% and leaves contained 0.44–5.44% [31]. Each country has different chemotypes of tansy. Most European countries have a dominant β-thujone type [43], but it is very individual. The Lithuanian study revealed that there was a 1,8-cineole chemotype [44]. In addition, the Estonian research showed that several chemotypes can be present in one country, depending on the location; the Harju district has trans-chrysanthenone, but Tartu has the β-thujone type [43]. Considering that the Latvian *T.vulgare* contains a relatively low concentration of thujone, it is unlikely that it will be responsible for the in vitro antiparasitic effect against Trichostrongylidae in this case.

In view of all the above mentioned, in order to evaluate and prove the effectiveness of the extract, it is necessary to carry out local research. In this study, extracts of *T.vulgare* growing in Latvia had an anthelmintic effect on Trichostrongylidae nematodes in sheep. There are several other studies that also demonstrate the effects of tansy extract on different parasite species and stages. These studies used ethanol and aqueous extract from aerial parts of *T.vulgare*; the strongest ovicidal effect was on donkey’s strongyles, where, at 31.25 mg/mL, 100% inhibition was demonstrated [45]. The lower effect was found for *A. suum* (76.57%) [46]. In our study, the highest percentage of egg inhibition was with the 200 mg/mL concentration of 50% acetone *T.vulgare* leaf extract (95.8%). In these researches were used different extract solvents, concentrations, and parasite species. However, all these solvents are polar and they have a good ovicidal effect on parasite eggs [32,47]. Effects on eggs have been described for plant secondary metabolites in an acetone extract; they induce changes in the permeability of the eggshell, inhibit hatching enzymes, and create competition for hatching factor receptors [32]. This could explain why the acetone extract had the strongest effect on eggs in our study.

In other study, was used ethanol extract from the aerial part of *T.vulgare* and detected larval migration inhibition to Trichostrongylus colubriformis L_3_ larvae in sheep. The highest percentage of inhibition was 49.15% at 2000 µg/mL [46]. There was one different study using an aqueous extract which showed a high larvicidal effect on donkey`s strongyles 98.57% at 125 mg/mL [45]. In our study, a strong larvicidal effect (100%) was detected at in different concentrations. However, there is also a study showing that water infusions of the *T.vulgare* had no larvicidal effect on *Strongyloides papillosus* [48]. Maybe these differences in tansy extract larvicidal activity are caused by plant parts and solvents because the amount of plant secondary metabolites varies in different plant parts and the solvents can extract specific metabolites, especially phenolic compounds which can disturb larval development [34].

In this study, adulticide did not affect Trichostrongylidae nematodes, but other similar studies showed this effect for *Chabertia ovina* [49], where ethanol extract of the aerial part of *T.vulgare* was used. Adulticide’s effect is not immediate because after 6 h there was no effect, but after 24 h the extract showed a strong effect on adult parasites [49]. A similar effect was observed in one in vivo study where the effect of the plant extract was observed only after 10 days [39]. On the other hand, the strongest adulticidal effect was observed for *T.vulgare* hydroalcoholic and essential oil on *Schistosoma mansoni* [40]. Both extracts proved to be 100% effective, but the essential oil reached 100% faster at 24 h than the hydroalcoholic at 72 h. Perhaps, when using tansy extract for adult stages of parasites, the effect is delayed because it takes time for the plant active compounds to be absorbed and this causes the tegumental alteration that was observed in this study [40]. *T. vulgare* extract not only has a nematocidal effect but also affects other parasite types, such as the protozoa *Giardia muris* [50] and the tapeworm *Echinococcus granulosus* protoscoleces [51]. Hydroalcoholic tincture and aqueous extract of *T.vulgare* flowers were used in both cases. The giardicidal effect can be explained by the content of polyphenols (gallic, ellagic, caffeic, p-coumaric, vanillic acids, and quercetin) [50], but what exactly causes the effect on *E.granulosus* was not described.

No other studies were found that used acetone as the solvent for *T.vulgare* extract. In our study, we used this solvent and there is a study that compared total phenolic compounds in ethanol and acetone extracts; acetone extracts contain more of these components than ethanol [52]. If compare the types of extract used in our study with this study, the acetone 30% flower extract contains the most phenolics. It was the only extract that had no antibacterial effect [52]. This was also proven in our research as this extract showed weaker ovicidal and larvicidal activity. This means that a high amount of total phenols does not yet prove the effectiveness of the extract; perhaps more important are the combinations of components that can influence the anthelmintic properties. This was also shown by a study on *Cooperia punctate*, where the combination of quercetin and rutin does not show an effect but a combination of quercetin, rutin, caffeic acid, and coumarin worked synergistically and had an effect on egg hatching and larval exsheathment [34].

Plant secondary metabolites not only have a positive effect but also a negative one for animals and parasites. Thujone with neurotoxicity, genotoxic [41,42] ∝-thujone, has stronger neurotoxicity than β-thujone, and both can cause acute and chronic toxicity, which is manifested by seizures of varying degrees [42]. Other research showed that *T.vulgare* leaf aqueous extract was quite safe for rodents [53]. Condensed tannins do not have as negative an effect as thujone, but also can reduce feed intake, fiber digestibility, and animal performance [36], change the concentration of acids and microorganisms in the rumen, and reduce the number of protozoa, especially *Dasythricha*, *Entodinium* and *Diplodinium* for sheep [54]. In this case, the dosage is important. If it is exceeded, then these negative aspects appear. Plant secondary metabolites do not have a positive effect on parasites, but only on their host. They can initiate the death of adult parasites in different ways. Terpenes induce intestinal damage; a similar effect is also observed for tannins [55]. Others affect energy metabolism; flavonoids block the phosphorylation reaction but alkaloids inhibit acetylcholine receptors and suppress glucose uptake [55]. Some metabolites directly initiate paralysis or death of the parasites; glycosides suggest a disturbance of sodium and potassium ions transportation, saponins inhibit acetylcholinesterase and non-protein amino acids initiate paralysis directly [55].

Given these facts, when conducting this kind of research, it is important to determine the maximally effective concentration or lethal dose, which allows a comparison of the safety of the plant [14]. In this study, the median lethal concentration was determined. The extracts of tansy were observed to be more toxic to Trichostrongylidae larvae than eggs due to their much lower LC_50_ values. The 50% ethanol and acetone extract of tansy leaf had a stronger ovicidal effect, and the lowest LC_50_ values were also observed for them. On the other hand, in order to determine the toxicity of these extracts on sheep, in vitro studies on cell cultures should be carried out, and followed by in vivo studies in sheep, i.e., initially under controlled conditions and then an on-farm study.

This study showed the high anthelmintic activity of *T.vulgare* extracts against Trichostrongylidae in sheep and it might establish a basis for further studies focused on searching for an alternative method to control gastrointestinal nematodes in sheep.

## 5. Conclusions

*Tanacetum vulgare* L. has ovicidal and larvicidal activity against the gastrointestinal nematodes (Trichostrongylidae) of sheep. The strongest ovicidal activity 95.8% had 200 mg/mL concentration of 50% acetone extract but almost all extracts had 100% larvicidal activity. Evaluating both activities, the stronger anthelmintic effect has 50% leaf extracts. Future studies are required to determine the anthelmintic activity of these extracts in vivo.

## Figures and Tables

**Table 1 animals-13-02176-t001:** Ovicidal activity of the *Tanacetum vulgare* L. leaf extracts.

Extracts	Concentration of Extracts (mg/mL) (±SD)						LC_50_ *
	500	200	100	50	20	10	
Tansy ethanol 30%	89.3 (±9.3)	76.1 (±3.7)	61.5 (±13.0)	51.1 (±15.5)	4.9 (±66.5)	0.0	113.8
Tansy ethanol 50%	93.3 (±6.3)	89.6 (±2.5)	27.8 (±47.4)	48.3 (±25.5)	57.5 (±19.5)	36.7 (±21.3)	31.2
Tansy ethanol 70%	44.4 (±12.3)	44.7 (±20.7)	55.7 (±20.6)	29.5 (±13.5)	3.9 (±31.6)	0.0	213.5
Tansy acetone 30%	0.0	0.0	82.3 (±15.7)	40.3 (±12.7)	60.7 (±19.4)	53.3 (±10.4)	0.04
Tansy acetone 50%	92.3 (±13.3)	95.8 (±7.2)	80.0 (±17.6)	73.3 (±18.9)	66.7 (±20.0)	9.0 (±0.0)	26.3
Tansy acetone 70%	90.9 (±3.0)	82.2 (±21.4)	87.7 (±13.7)	38.8 (±2.0)	52.3 (±13.7)	64.4 (±17.3)	11.7

* Median lethal concentration (mg/mL).

**Table 2 animals-13-02176-t002:** Ovicidal activity of the *Tanacetum vulgare* L. flower extracts.

Extracts	Concentration of Extracts (mg/mL) (±SD)						LC_50_ *
	500	200	100	50	20	10	
Tansy ethanol 30%	16.8 (±49.5)	0.0	27.2 (±32.3)	50.9 (±46.0)	0.0	0.0	2366.8
Tansy ethanol 50%	77.6 (±20.9)	43.4 (±39.1)	68.4 (±7.9)	74.4 (±14.1)	0.0	81.8 (±1.4)	142.3
Tansy ethanol 70%	56.5 (±37.1)	37.9 (±41.9)	43.2 (±9.6)	0.0	15.6 (±38.9)	45.8 (±56.2)	3279.2
Tansy acetone 30%	33.6 (±67.9)	52.6 (±27.9)	0.0	42.9 (±64.4)	0.0	0.0	551.8
Tansy acetone 50%	91.5 (±4.0)	70.0 (±23.8)	84.3 (±19.0)	75.1 (±10.3)	55.5 (±25.5)	6.8 (±41.2)	37.4
Tansy acetone 70%	19.3 (±17.5)	0.0	0.0	0.0	0.0	0.0	17,426.3

* Median lethal concentration (mg/mL).

**Table 3 animals-13-02176-t003:** Larvicidal activity of the *Tanacetum vulgare* L. leaf extracts.

Extracts	Concentration of Extracts (mg/mL) (±SD)						LC_50_ *
	500	200	100	50	20	10	
Tansy ethanol 30%	100.0 (±0.0)	100.0 (±57.7)	100.0 (±0.0)	100.0 (±0.0)	96.4 (±9.6)	85.7 (±27.5)	1.26
Tansy ethanol 50%	100.0 (±57.7)	100.0 (±0.0)	100.0 (±57.7)	100.0 (±57.7)	100.0 (±57.7)	100.0 (±57.7)	-
Tansy ethanol 70%	100.0 (±0.0)	100.0 (±0.0)	100.0 (±0.0)	100.0 (±0.0)	100.0 (±0.0)	100.0 (±0.0)	-
Tansy acetone 30%	100.0 (±0.0)	-	100.0 (±0.0)	100.0 (±57.7)	100.0 (±0.0)	100.0 (±0.0)	-
Tansy acetone 50%	100.0 (±0.0)	100.0 (±0.0)	100.0 (±0.0)	100.0 (±0.0)	100.0 (±0.0)	100.0 (±0.0)	-
Tansy acetone 70%	100.0 (±0.0)	100.0 (±0.0)	100.0 (±0.0)	100.0 (±0.0)	100.0 (±57.7)	100.0 (±0.0)	-

* Median lethal concentration (mg/mL).

**Table 4 animals-13-02176-t004:** Larvicidal activity of the *Tanacetum vulgare* L. flower extracts.

Extracts	Concentration of Extracts (mg/mL) (±SD)						LC_50_ *
	500	200	100	50	20	10	
Tansy ethanol 30%	-	-	100.0 (±57.7)	100.0 (±57.7)	100.0 (±0.0)	100.0 (±57.7)	-
Tansy ethanol 50%	100.0 (±0.0)	100.0 (±57.7)	100.0 (±0.0)	100.0 (±0.0)	100.0 (±0.0)	100.0 (±57.7)	-
Tansy ethanol 70%	100.0 (±57.7)	100.0 (±0.0)	100.0 (±0.0)	100.0 (±0.0)	100.0 (±0.0)	100.0 (±0.0)	-
Tansy acetone 30%	-	100.0 (±57.7)	88.9 (±54.1)	33.3 (±57.7)	81.8 (±25.2)	77.8 (±28.9)	10.00
Tansy acetone 50%	100.0 (±57.7)	100.0 (±57.7)	100.0 (±0.0)	100.0 (±0.0)	100.0 (±0.0)	94.1 (±8.2)	0.02
Tansy acetone 70%	100.0 (±0.0)	100.0 (±0.0)	100.0 (±0.0)	100.0 (±0.0)	100.0 (±57.7)	97.3 (±5.2)	0.005

* Median lethal concentration (mg/mL).

## Data Availability

Not applicable.

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
