# Peer review of "Anthelmintic Activity of Tanacetum vulgare L. (Leaf and Flower) Extracts against Trichostrongylidae Nematodes in Sheep In Vitro"

_animals, 2023, doi:10.3390/ani13132176_

Round 1
Reviewer 1 Report
Phytotherapy is a promisingly alternative method to chemical therapy in the control of gastrointestinal nematodes in small ruminants, since the latter is facing resistance, residue and environment pollution. The study demonstrate all tansy extracts with ethanol and acetone had ovicidal and larvicidal activity against Trichostrongylidae in sheep, showing the potential of application in control of nematodes in small ruminants. The research is interesting although it’s preliminary one. Some comments or suggestions are below.
1) L106-119 Why the nonpolar solvent was not tried to extract the tansy?
2) L118 Which solvent was used to dilute the extract?
3) L121-133 Did the sheep naturally infect Trichostrongylidae by mixed species? If so, were the species same or not between different batches of eggs? Did this factor affect the efficacy of the extract against eggs and larvae?
4) Add SD to the values in table 1 to 4.
5) In table 1, it’s strange that high concentration of tansy acetone 30% show no ovicidal activity but low one show high ovicidal activity. Similar results were also observed in other tables. Please give more explanation to these “strange” results.
6) In the evaluation of larvicidal activity of the extract, why such high concentrations of the extract were used?
7) I’d like to suggest authors to pay attention to the identification of active ingredients in the extract in further study.
1) L113-114 Replace the word “and” between “acetonic” and “ethanolic” with “or”.
2) In four tables, is the phrase “concentration of extracts” more accurate than “dilution of extracts” ?
Author Response
Dear Reviewer,
Thanks for your review! We have carefully revised the manuscript taking into account all your suggestions and comments. Please see the attachment.
We would like to thank you again for taking the time to review our manuscript.
Sincerely,
Alīna Kļaviņa

Reviewer 2 Report
In general, the work is well written and may be accepted by the journal after reviewing the discussions. I suggest that the discussions be rewritten comparing the results obtained with data from the literature.

Author Response

(The authors gave the same response as above.)

Reviewer 3 Report
Anthelmintic Activity of Tanacetum vulgare L. (Leaf and Flower) Extracts against Trichostrongylidae Nematodes in Sheep in Vitro
Many Greetings! Thank you for giving an opportunity for reviewing this manuscript.
The Authors have performed quite interesting work towards the scientific society and controlling Nematodes in Sheep using natural therapy. As a reviewer I am recommending acceptance with major revision only if the authors revise the manuscript in accordance with the comments below.
General comments:
Abstract
Line 36 - Concentration and dilution missing for larvae inhibition
The Botanical name Tanacetum vulgare is nowhere mentioned in the abstract
Introduction
- Line 115: The temperature and duration of rotary evaporation could be provided.
2. Line 117: It could be mentioned how the dilutions were prepared from the extracts (e.g., serial dilution method).
2. Materials and Methods
2.1. Ethical Approval reference no could be add
2.2. Plant Material Preparation:
Authors should be reveal what chemical components present in the Tanacetum vulgare plant extracts (add GC-MS analysis of plant extract); and discuss with previous biological reports of your plant compounds.
2.3.1. Parasites and Recovery/Obtaining of Eggs
1. Line 122: Specify the number of sheep from which fecal samples were collected.
2. Line 123: Provide details about the Trichostrongylidae species identified in the fecal samples.
2.3.2. Egg Hatching Test
1. Line 157: After incubation, Lugol’s iodine solution- Specify the concentration or volume of Lugol's iodine solution used.
2. Line 158: Specify the specific stages of larvae counted (L1, L2, L3).
2.3.3 Larval Development Test
1.Line 149: diluted with a solution of Amphotericin B (What is the purpose of using this?)
3. Results
1. Line 193: The leaf extracts had lower LC50 values than the flower extracts. The lowest LC50 – 50 must be in subscript.
2. Line 209: The lowest LC50 value of 0.005 mg/mL was for the 70% acetone extract of tansy flowers – LC50
The author has well planned and executed their findings. In overview, there are several typographical errors have been found. In particular, the many places genus and species name not in italic. There are several grammatical errors are found and the author has to consider briefly.
Author Response

(The authors gave the same response as above.)

Round 2
Reviewer 3 Report
The Authors carryout all the correction now the manuscript was suitable for publication